# The impact of cognitive impairment and low muscle mass on all-cause mortality among older adults in China: An empirical analysis based on CLHLS cohort data

Huixia Cao[1,2]☯, Jiefen Ou[1]☯*, Manzhi Gao[2], Yingying Zhong[2]

**1** Department of Nursing, The Sixth Affiliated Hospital of South China University of Technology (Nanhai District People's Hospital of Foshan), Foshan, Guangdong, China, **2** School of Nursing, Guangdong Pharmaceutical University, Guangzhou, Guangdong, China

☯ These authors contributed equally to this work.
* ojf2854529676@163.com

## Abstract

### Background

Cognitive impairment and low muscle mass are prevalent conditions in later life, and each has been independently associated with adverse health outcomes. Both conditions are heterogeneous and may arise from multiple underlying etiologies. However, evidence regarding their combined association with all-cause mortality remains limited, particularly among older adults in China. Therefore, this study aimed to investigate the independent and joint associations of cognitive impairment and low muscle mass with all-cause mortality using data from a large prospective cohort.

### Methods

Data from the China Longitudinal Healthy Longevity Survey (CLHLS) collected between 2011 and 2018 were analyzed. Kaplan–Meier survival analyses and log-rank tests were conducted, and Cox proportional hazards models were applied to estimate hazard ratios (HRs) and corresponding 95% confidence intervals (CIs) for all-cause mortality. Subgroup and sensitivity analyses were additionally performed to evaluate the robustness of the findings.

### Results

A total of 5,625 participants were included and classified into four groups: neither cognitive impairment (CI) nor low muscle mass (LMM) (CI−/LMM−, n = 1,698; 30.2%), low muscle mass only (CI−/LMM+, n = 2,795; 49.7%), cognitive impairment only (CI+/LMM−, n = 341; 6.1%), and both cognitive impairment and low muscle mass (CI+/LMM+, n = 791; 14.1%). Kaplan–Meier survival analyses and multivariable Cox proportional hazards models demonstrated that participants in the CI+/

**Data availability statement:** Publicly available datasets were analyzed in this study. This data can be found at: The data that support the findings of this study are openly available at: https://opendata.pku.edu.cn/dataverse/CHADS.

**Funding:** The author(s) received no specific funding for this work.

**Competing interests:** The authors have declared that no competing interests exist.

LMM+ group had the highest risk of all-cause mortality (p < 0.001). Compared with the CI−/LMM− group, the adjusted hazard ratios (HRs) were 1.29 (95% CI: 1.01–1.51) for the CI−/LMM+ group, 1.92 (95% CI: 1.50–2.45) for the CI+/LMM− group, and 2.54 (95% CI: 2.11–3.05) for the CI+/LMM+ group. Sensitivity analyses confirmed the robustness of the main findings, revealing a significant increasing trend in mortality risk across the four groups (trend p < 0.001). In subgroup analyses, the CI+/LMM+ group was significantly associated with an elevated risk of all-cause mortality among women and participants aged <80 years.

## Conclusions

The coexistence of cognitive impairment and low muscle mass was associated with a substantially increased risk of all-cause mortality, particularly among women and individuals younger than 80 years. The concurrent presence of these conditions may help identify a subgroup of older adults at heightened risk of adverse outcomes, underscoring the importance of comprehensive geriatric assessment and enhanced clinical surveillance in aging populations.

## Introduction

With the accelerated aging of the global population, cognitive impairment and low muscle mass have emerged as two major health challenges among older adults. Cognitive impairment, characterized by declines in memory, executive function, and social adaptability, is common among community-dwelling older adults, with prevalence increasing markedly with age [1]. Epidemiological evidence indicates that cognitive impairment, as assessed using standardized instruments, is associated with a range of adverse outcomes, including functional decline, loss of independence, and increased all-cause mortality in later life [2,3]. Similarly, low muscle mass is highly prevalent in aging populations and reflects age-related changes in body composition, physical exercise, nutritional status, and overall health [4]. Population-based studies consistently demonstrate that low muscle mass is independently associated with disability, frailty, and an elevated risk of mortality among older adults [5]. Emerging evidence further suggests that cognitive impairment and low muscle mass frequently coexist in older individuals, such that those presenting with one condition are more likely to exhibit the other. This coexistence implies that these two common aging-related conditions may be interrelated and jointly contribute to adverse health outcomes [6], thereby warranting closer examination of their combined impact.

Building on this observed coexistence, the interaction between cognitive impairment and low muscle mass has garnered increasing attention in aging research, as both conditions are highly prevalent among older adults and are strongly associated with adverse health outcomes. Epidemiological studies indicate that mild cognitive impairment (MCI) affects approximately 15% to over 30% of community-dwelling adults aged 60 years and older [2]. Compared with cognitively normal individuals, older adults experiencing accelerated cognitive decline have a 24% higher risk of

mortality [7]. Similarly, low muscle mass—commonly assessed using anthropometric or body composition measures—is highly prevalent among older adults, affecting 45.2% of the Chinese population [4], and has been associated with a 57% increase in the risk of all-cause mortality in meta-analyses [8]. Longitudinal evidence further suggests that older adults with low muscle mass are more likely to experience subsequent cognitive decline than those with preserved muscle mass [9]. In addition, population-based observational studies have demonstrated a significant association between cognitive impairment and low muscle mass in older adults, with longitudinal evidence suggesting that reduced skeletal muscle mass is associated with an increased risk of subsequent cognitive impairment [10]. Taken together, these findings suggest that the co-occurrence of cognitive impairment and low muscle mass may identify a subgroup of older adults at particularly high risk of adverse health outcomes, underscoring the importance of examining their joint association with mortality. Despite the accumulating evidence on the prevalence and adverse consequences of cognitive impairment and low muscle mass, existing research has predominantly focused on single-disease phenotypes, with limited quantitative assessment of the mortality risk associated with their comorbidity. Moreover, these associations have not been adequately examined in older Chinese populations. Using 7-year follow-up data from the China Longitudinal Healthy Longevity Survey (CLHLS), the present study systematically investigates the joint effects of cognitive impairment and low muscle mass on all-cause mortality among older adults. The findings may help identify individuals at elevated risk of adverse outcomes and provide population-level evidence to inform geriatric risk stratification and health resource planning. Ultimately, this study aims to contribute robust epidemiological evidence to support future research on comprehensive health assessment and management in aging populations.

## Materials and methods

### Study design and population

The China Longitudinal Healthy Longevity Survey (CLHLS) is an ongoing prospective cohort study targeting older adults aged 60 years and above. The survey collects comprehensive information on participants' demographic characteristics, socioeconomic status, health-related behaviors, and physical health conditions. Detailed descriptions of the CLHLS design and procedures have been published elsewhere [11,12]. The study protocol was approved by the Biomedical Ethics Committee of Peking University, Beijing, China (IRB00001052–13074). Written informed consent was obtained from all participants or their legal representatives. The inclusion and exclusion criteria are illustrated in Fig 1. Participants were eligible for inclusion if they were enrolled in the CLHLS between 2011 and 2018, aged ≥65 years, had complete data on cognitive function, muscle mass, and relevant covariates, and had no diagnosis of dementia. Participants were excluded if they met any of the following criteria: (a) age < 65 years; (b) absence of cognitive assessment data; (c) missing data on muscle mass; (d) a diagnosis of dementia; or (e) missing data on key covariates (e.g., smoking, drinking). After applying these criteria, 5,625 individuals with complete data on core variables were retained from the initial 10,891 baseline participants and included in the final analytical sample.

### Assessment of cognitive impairment

Cognitive function was assessed using the Chinese version of the Mini-Mental State Examination (MMSE), a widely used instrument for evaluating global cognitive status. The Chinese MMSE was adapted from the original international version to better reflect the cultural and socioeconomic context of older adults in China [13]. The instrument comprises 24 items covering five cognitive domains: general abilities (6 items), response abilities (3 items), attention and calculation abilities (6 items), recall (3 items), and language, comprehension, and self-coordination abilities (6 items).The total MMSE score is calculated as the sum of all item scores, ranging from 0 to 30 points, with higher scores indicating better cognitive function. To account for heterogeneity in educational attainment, education-specific cutoff points were applied to define cognitive impairment: a score of <18 for illiterate individuals (0 years of schooling), < 21 for those with primary school level education (1–6 years of schooling), and <25 for participants with secondary school education or above (>6 years of

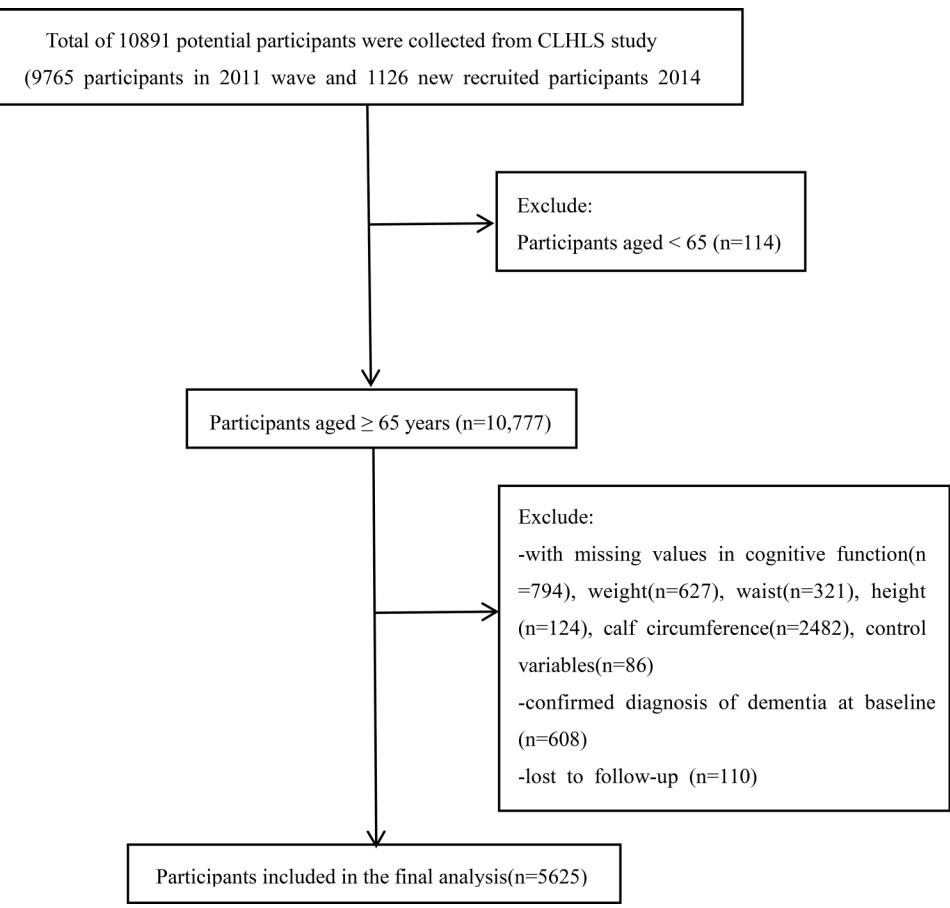

**Fig 1. Flowchart for participant screening and exclusion criteria.**

schooling) [14]. Participants scoring below the corresponding education-specific cutoff were classified as having cognitive impairment.

## Assessment of low muscle mass

Muscle mass generally refers to appendicular skeletal muscle mass (ASM), which is commonly measured using dual-energy X-ray absorptiometry (DXA) or bioelectrical impedance analysis. However, due to limitations in the available dataset, direct measurements of muscle mass were not accessible. Therefore, muscle mass in this study was estimated using a validated anthropometric prediction equation originally developed and validated in Chinese community-dwelling older adults [15]. This equation has been widely applied in recent nationwide cohort studies of low muscle mass among Chinese older adults [16]. In accordance with the Asian Obesity Guidelines (AOGC) [17], calf circumference was adopted as the primary anthropometric indicator for community-based screening of muscle mass at baseline. Calf circumference data were obtained from the China Longitudinal Healthy Longevity Survey (CLHLS), where standardized anthropometric measurements were conducted by trained interviewers. During the assessment, participants were asked to sit with the sole of their right foot flat on the ground, with the right calf and thigh positioned at a 90-degree angle. A flexible measuring tape was used to measure the circumference of the right calf at its widest point, and the value was recorded to the nearest integer. This approach was applied to estimate skeletal muscle mass for epidemiological analyses rather than to establish

a clinical diagnosis of sarcopenia. Given that different obesity phenotypes may influence calf circumference measurements [18], a multivariable regression equation incorporating key anthropometric variables was used to predict ASM. The prediction equation was as follows: ASM (kg) = 2.955 × sex (male = 1, female = 0) + 0.255 × weight (kg) − 0.130 × waist circumference (cm) + 0.308 × calf circumference (cm) + 0.081 × height (cm) − 11.897 (adjusted $R^2$ = 0.94, standard error of the estimate = 1.2 kg). Low muscle mass (LMM) was defined using a height-squared normalization method (ASM/height²), with cutoff values of <7.0 kg/m² for males and <5.7 kg/m² for females.

## Study outcomes

The primary outcome of this study was all-cause mortality. The analysis included two baseline cohorts from the China Longitudinal Healthy Longevity Survey (CLHLS), comprising participants initially interviewed in 2011 and those newly recruited in 2014. All participants were followed through 2018. Survival time was defined as the interval from the baseline interview date (2011 or 2014) to the date of death or the date of the most recent confirmed contact. Information on survival status and date of death was obtained during the 2018 follow-up survey, based on reports from close relatives or community health workers, in accordance with standard CLHLS procedures. Participants who were not reported as deceased but could not be contacted during follow-up were classified as lost to follow-up; a total of 110 participants met this criterion and were excluded prior to the final analytic sample. Participants who were alive or lost to follow-up were censored at the date of their most recent confirmed contact, with December 30, 2018, designated as the administrative censoring date for the final follow-up.

## Assessment of potential confounders

Potential confounders included demographic characteristics, health-related behaviors, health conditions, and physical functioning. Demographic variables comprised age, place of residence (urban, town, or rural), sex (male or female), marital status (married, divorced, widowed, or unmarried), educational attainment (illiterate [0 years], primary school [1–6 years of schooling], or secondary school and above [>6 years of schooling]), and household average income (low, medium, or high). Health-related behaviors were assessed using body mass index (BMI), categorized as <18.5 kg/m², 18.5–23.9 kg/m², 24.0–27.9 kg/m², and ≥28.0 kg/m² [19], physical exercise, drinking status, and smoking status. Physical exercise (yes or no) was measured by asking "do you take exercise regularly in the present?". Smoking (yes or no) was assessed by asking "do you smoke at present?" and drinking (yes or no) was assessed by asking "do you drink at present?". Health conditions were defined based on self-reported physician diagnoses of heart disease (coronary heart disease, myocardial infarction, heart valve disease, and other cardiac conditions), hypertension, diabetes, stroke (various acute cerebrovascular events such as cerebral infarction and cerebral hemorrhage), respiratory diseases (bronchitis, emphysema, asthma, or pneumonia), arthritis, cancer, and sensory impairment (visual or hearing impairment), with all variables coded as binary (yes/no). Physical functioning was assessed using standardized questionnaires for Basic Activities of Daily Living (BADL) and Instrumental Activities of Daily Living (IADL). The BADL scale includes six fundamental self-care activities, whereas the IADL scale assesses eight more complex tasks required for independent living. Each item was rated on a three-point scale (1 = fully independent, 2 = partially dependent, 3 = fully dependent). BADL impairment was defined as the inability to independently perform at least one of the six BADL tasks, and IADL limitation was defined as the inability to independently complete at least one of the eight IADL tasks. BADL and IADL impairment were used as proxies for functional dependence and care needs in community-dwelling older adults [20].

## Statistical analysis

Descriptive statistics were used to summarize baseline characteristics. Differences in survival probabilities were examined using Kaplan–Meier (K–M) survival curves and the log-rank test. Multivariable Cox proportional hazards regression models were constructed to evaluate the associations of cognitive impairment and low muscle mass with all-cause mortality.

Four sequential models were fitted: Model 1 was unadjusted; Model 2 was adjusted for sex, place of residence, marital status, educational attainment, and household average income; Model 3 was further adjusted for physical exercise, smoking status, alcohol consumption, and body mass index; and Model 4 additionally included heart disease, hypertension, diabetes, stroke, respiratory disease, arthritis, cancer, sensory impairment, and functional status (BADL and IADL).The proportional hazards (PH) assumption was assessed using Schoenfeld residuals. Subgroup analyses and interaction tests were conducted based on the fully adjusted model (Model 4) to explore potential effect modification across predefined subgroups. Sensitivity analyses were performed to evaluate the robustness of the findings by excluding participants with a history of stroke or heart disease, as well as those who died within the first two years of follow-up.All statistical analyses were performed using R software (version 4.4.3). A two-sided p value < 0.05 was considered statistically significant.

## Results

### Demographic characteristics of participants

Among the 5,625 participants included in the analysis, 1,698 (30.2%) had neither cognitive impairment nor low muscle mass (CI−/LMM−), 2,795 (49.7%) had low muscle mass only (CI−/LMM+), 341 (6.1%) had cognitive impairment only (CI+/LMM−), and 791 (14.1%) had both cognitive impairment and low muscle mass (CI+/LMM+).Baseline characteristics of the four groups are presented in Table 1. Significant differences were observed across groups for multiple variables (p<0.001). Compared with the other groups, participants in the CI+/LMM+ group were more likely to be older, reside in rural areas, be male, widowed, and illiterate, and to have a lower body mass index, lower levels of physical exercise, impairment in basic and instrumental activities of daily living (BADL and IADL), sensory impairment, and a higher proportion of all-cause mortality (all p<0.001) (Table 1).

### Association between cognitive impairment and low muscle mass

Logistic regression analysis showed that cognitive impairment was significantly associated with a higher likelihood of low muscle mass (odds ratio [OR] = 2.52, 95% confidence interval [CI]: 2.13–3.01, p<0.001). This association remained statistically significant after adjustment for potential confounders in the multivariable logistic regression model (adjusted odds ratio [aOR] = 1.45, 95% CI: 1.15–1.69, p<0.001) (Table 2).

### Impact of cognitive impairment and low muscle mass on all-cause mortality

Using longitudinal follow-up data from 5,625 participants in the China Longitudinal Healthy Longevity Survey (CLHLS), the associations of cognitive impairment (CI) and low muscle mass (LMM) with all-cause mortality were examined using Kaplan–Meier survival analysis and Cox proportional hazards regression models. Kaplan–Meier survival curves (Fig 2) demonstrated significant differences in overall survival across the four groups during a median follow-up of 85 months (log-rank p<0.001). Participants in the CI−/LMM− group exhibited the highest survival probability, whereas those in the CI+/LMM+ group had the lowest survival probability.

In the unadjusted model (Model 1), compared with the CI−/LMM− group, the risk of all-cause mortality was significantly higher in the CI−/LMM+ group (hazard ratio [HR] = 1.87, 95% confidence interval [CI]: 1.63–2.14), the CI+/LMM− group (HR = 3.83, 95% CI: 3.03–4.84), and the CI+/LMM+ group (HR = 5.93, 95% CI: 5.12–6.86) (all p<0.001). Potential confounders were selected based on univariate analyses and clinical relevance. Multicollinearity among covariates was assessed using variance inflation factors (VIFs), which ranged from 1.001 to 1.256 (S1 Table in S1 File), indicating negligible multicollinearity and minimal risk of inflated standard errors or biased hazard ratio estimates. After adjustment for all confounders in the fully adjusted model (Model 4), the associations remained statistically significant. Compared with the CI−/LMM− group, participants in the CI−/LMM+ group had a 29% higher risk of all-cause mortality (HR = 1.29, 95% CI: 1.10–1.51), those in the CI+/LMM− group had a 92% higher risk (HR = 1.92, 95% CI: 1.50–2.45), and those in the CI+/LMM+ group had the highest risk (HR = 2.54, 95% CI: 2.11–3.05) (all p<0.001). Across all models, a significant

**Table 1.** Baseline characteristics of the study subjects.

| | CI-/LMM-<br>(n = 1698) | CI-/LMM+<br>(n = 2795) | CI+/LMM-<br>(n = 341) | CI+/LMM+<br>(n = 791) | P value |
|---|---|---|---|---|---|
| Age(years) | 79.64 ± 8.53 | 84.99 ± 9.64 | 88.19 ± 9.57 | 95.74 ± 8.27 | <0.001 |
| Residence,% | | | | | <0.001 |
| Urban | 314 (18.5) | 284 (10.2) | 47 (13.8) | 73 (9.2) | |
| Town | 510 (30.0) | 867 (31.0) | 90 (26.4) | 187 (23.6) | |
| Rural | 874 (51.5) | 1644 (58.8) | 204 (59.8) | 531 (67.1) | |
| Gender, Male,% | 1068 (62.9) | 1211 (43.3) | 163 (47.8) | 195 (24.7) | <0.001 |
| Marriage,% | | | | | <0.001 |
| Married | 982 (57.8) | 1054 (37.7) | 102 (29.9) | 94 (11.9) | |
| Divorced | 33 (1.9) | 68 (2.4) | 6 (1.8) | 6 (0.8) | |
| Widowed | 663 (39.0) | 1643 (58.8) | 229 (67.2) | 684 (86.5) | |
| Unmarried | 20 (1.2) | 30 (1.1) | 4 (1.2) | 7 (0.9) | |
| Education,% | | | | | <0.001 |
| Illiterate | 659 (38.8) | 1668 (59.7) | 203 (59.5) | 640 (80.9) | |
| Primary school | 709 (41.8) | 860 (30.8) | 105 (30.8) | 122 (15.4) | |
| Secondary school or higher | 330 (19.4) | 267 (9.6) | 33 (9.7) | 29 (3.7) | |
| Average household income (CNY),% | | | | | <0.001 |
| Low | 542 (31.9) | 946 (33.8) | 107 (31.4) | 259 (32.7) | |
| Medium | 535 (31.5) | 1024 (36.6) | 125 (36.7) | 321 (40.6) | |
| High | 621 (36.6) | 825 (29.5) | 109 (32.0) | 211 (26.7) | |
| BMI,% | | | | | <0.001 |
| <18.5 | 29 (1.7) | 750 (26.8) | 48 (14.1) | 308 (38.9) | |
| 18.5–23.9 | 658 (38.8) | 1827 (65.4) | 173 (50.7) | 423 (53.5) | |
| 24–27.9 | 717 (42.2) | 209 (7.5) | 82 (24.0) | 53 (6.7) | |
| ≥28 | 294 (17.3) | 9 (0.3) | 38 (11.1) | 7 (0.9) | |
| Physical exercise,% | 644 (37.9) | 784 (28.1) | 80 (23.5) | 70 (8.8) | <0.001 |
| Drinking,% | 348 (20.5) | 422 (15.1) | 49 (14.4) | 74 (9.4) | <0.001 |
| Smoking,% | 356 (21.0) | 512 (18.3) | 42 (12.3) | 78 (9.9) | <0.001 |
| Heart diseases,% | 262 (15.4) | 323 (11.6) | 55 (16.1) | 66 (8.3) | <0.001 |
| Hypertension,% | 736 (43.3) | 863 (30.9) | 112 (32.8) | 170 (21.5) | <0.001 |
| Diabetes,% | 145 (8.5) | 115 (4.1) | 23 (6.7) | 12 (1.5) | <0.001 |
| Stroke,% | 183 (10.8) | 153 (5.5) | 44 (12.9) | 62 (7.8) | <0.001 |
| BADL disability,% | 58 (3.4) | 135 (4.8) | 60 (17.6) | 231 (29.2) | <0.001 |
| IADL disability,% | 299 (17.6) | 865 (30.9) | 185 (54.3) | 583 (73.7) | <0.001 |
| Respiratory disease,% | 175 (10.3) | 319 (11.4) | 32 (9.4) | 82 (10.4) | 0.713 |
| Arthritis,% | 238 (14.0) | 394 (14.1) | 48 (14.1) | 101 (12.8) | 0.778 |
| Cancer,% | 9 (0.5) | 26 (0.9) | 1 (0.3) | 2 (0.3) | 0.211 |
| Sensory impairment,% | 653 (38.5) | 1405 (50.3) | 236 (69.2) | 679 (85.8) | <0.001 |
| Mortality,% | 285 (16.8) | 775 (27.7) | 162 (47.5) | 500 (63.2) | <0.001 |

Data are shown as the mean ± SD or frequency (percentage). SD, standard deviation; BMI, Body mass index; CI, Cognitive impairment; LMM, Low muscle mass; CNY, Chinese Yuan; BADL, Basic activity of daily living; IADL, Instrumental activity of daily living.

increasing trend in mortality risk was observed across the four groups (all trend p values <0.001). Furthermore, across all four Cox regression models, cognitive impairment was consistently associated with a higher risk of all-cause mortality than low muscle mass. In the fully adjusted model, cognitive impairment was associated with a nearly twofold increase in

**Table 2. The association between cognitive impairment and low muscle mass.**

| Categories | Crude model | | Adjusted model | |
|---|---|---|---|---|
| | OR and 95%CI | P value | OR and 95%CI | P value |
| Non-low muscle mass | Ref. | | Ref. | |
| Low muscle mass | 2.52 (2.13,3.01) | <0.001 | 1.45 (1.15,1.69) | <0.001 |

Model adjusted for age, residence, gender, marriage, education, average household income, physical exercise, drinking, smoking, BMI, heart disease, hypertension, diabetes, stroke, BADL disability, IADL disability, respiratory diseases, arthritis, cancer, sensory impairment. BMI, Body mass index; BADL, basic activity of daily living; IADL, instrumental activity of daily living; ref, reference; CI, confidence interval; OR, odds ratio.

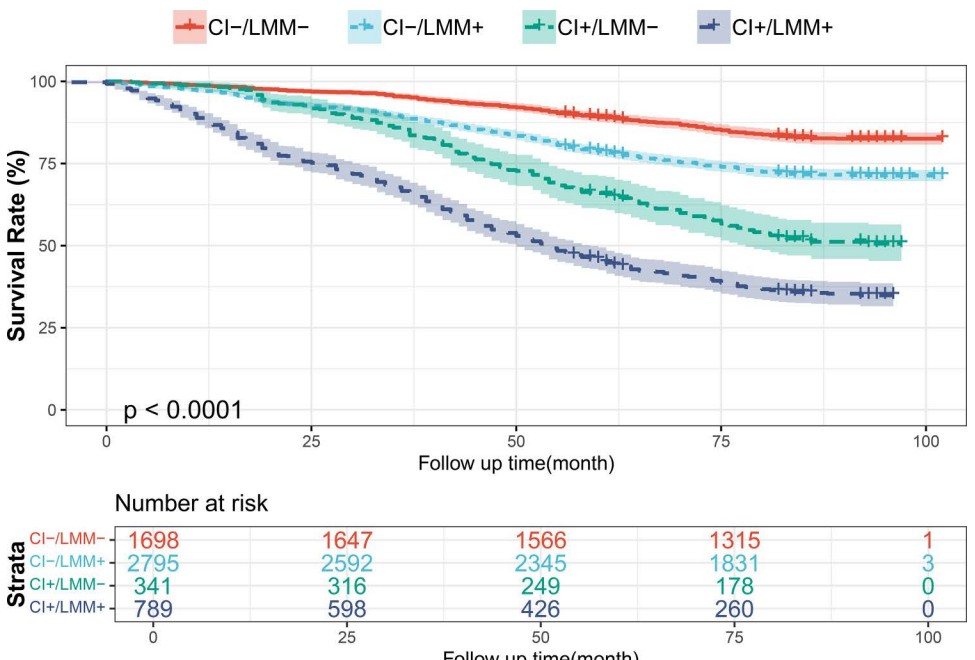

**Fig 2. The Kaplan-Meier survival curves for cognitive impairment and low muscle mass associated with risk of all-cause mortality.**

mortality risk (HR = 1.97, 95% CI: 1.75–2.21), whereas low muscle mass was associated with a more modest but significant increase in risk (HR = 1.31, 95% CI: 1.14–1.51) (both p < 0.001) (Table 3).

## Sensitivity analysis

Sensitivity analyses were conducted to assess the robustness of the main findings by excluding participants with a history of stroke, those with a history of heart disease, and those who died within the first two years of follow-up.

First, after excluding participants with a history of stroke (S2 Table in S1 File), the associations between cognitive impairment, low muscle mass, and all-cause mortality remained statistically significant. Compared with the CI−/LMM− group, the CI−/LMM+ group had a higher risk of all-cause mortality (hazard ratio [HR] = 1.20, 95% confidence interval [CI]: 1.02–1.42), as did the CI+/LMM− group (HR = 1.68, 95% CI: 1.29–2.20) and the CI+/LMM+ group, which exhibited the highest risk (HR = 2.38, 95% CI: 1.97–2.88) (all p < 0.001). A significant increasing trend in mortality risk across the four groups was observed (trend p < 0.001).

**Table 3. Hazard ratios for all-cause mortality according to the presence of cognitive impairment and low muscle mass.**

| Mortality outcomes | Model 1 | Model 2 | Model 3 | Model 4 | P-for-trend |
|---|---|---|---|---|---|
| | HR (95% CI) | HR (95% CI) | HR (95%CI) | HR (95% CI) | |
| All-cause mortality | | | | | |
| CI-/LMM- | Ref. | | | | |
| CI-/LMM+ | 1.87 (1.63, 2.14) | 1.46 (1.27, 1.68) | 1.32 (1.13, 1.55) | 1.29 (1.10, 1.51) | <0.001 |
| CI+/LMM- | 3.83 (3.03, 4.84) | 2.67 (2.11, 3.39) | 2.51 (1.97, 3.18) | 1.92 (1.50, 2.45) | <0.001 |
| CI+/LMM+ | 5.93 (5.12, 6.86) | 3.88 (3.31, 4.53) | 3.28 (2.75, 3.91) | 2.54 (2.11, 3.05) | <0.001 |
| Cognitive impairment* | 3.56 (3.22, 3.93) | 2.70 (2.43, 3.00) | 2.50 (2.25, 2.78) | 1.97 (1.75, 2.21) | <0.001 |
| Low musclemass* | 2.05 (1.83, 2.29) | 1.52 (1.35, 1.71) | 1.36 (1.18, 1.57) | 1.31 (1.14, 1.51) | <0.001 |

Model 1: Unadjusted.

Model 2: Adjusted for age, residence, gender, marriage, education, average household income.

Model 3: Adjusted for age, residence, gender, marriage, education, average household income,physical exercise, drinking, smoking, BMI.

Model 4: Adjusted for age, residence, gender, marriage, education, average household income, physical exercise, drinking, smoking, BMI, heart disease, hypertension, diabetes, stroke, BADL disability, IADL disability, respiratory diseases, arthritis, cancer, sensory impairment. CI, cognitive impairment; LMM, low muscle mass; BMI, Body mass index; HR, hazard ratio; ref, reference. *No cognitive impairment was set as the reference group; *No low muscle mass was set as the reference group.

Second, exclusion of participants with a history of heart disease (S3 Table in S1 File) yielded consistent results. Relative to the CI−/LMM− group, the CI−/LMM+ group showed an increased risk of mortality (HR = 1.31, 95% CI: 1.11–1.56), the CI+/LMM− group also demonstrated an elevated risk (HR = 1.82, 95% CI: 1.39–2.38), and the CI+/LMM+ group again had the highest risk (HR = 2.56, 95% CI: 2.10–3.11) (all $p < 0.001$). The increasing trend across groups remained significant (trend p < 0.001).

Finally, after excluding participants who died within the first two years of follow-up (S4 Table in S1 File), the pattern of associations persisted. Compared with the CI−/LMM− group, mortality risk was higher in the CI−/LMM+ group (HR = 1.31, 95% CI: 1.09–1.56), the CI+/LMM− group (HR = 2.51, 95% CI: 1.92–3.29), and the CI+/LMM+ group, which continued to exhibit the highest risk (HR = 2.70, 95% CI: 2.18–3.34) (all p < 0.001). A significant increasing trend in mortality risk across the four groups was again observed (trend p < 0.001).

Overall, all sensitivity analyses consistently confirmed that participants with coexisting cognitive impairment and low muscle mass (CI+/LMM+) had the highest risk of all-cause mortality (HR range: 2.38–2.70), and that mortality risk increased progressively across the four groups.

## Subgroup analysis

Stratified analyses by sex and age revealed heterogeneity in the associations of cognitive impairment (CI) and low muscle mass (LMM) with all-cause mortality across subgroups (Table 4). Among men, compared with the CI−/LMM− group, participants in the CI+/LMM+ group had a significantly higher risk of all-cause mortality (hazard ratio [HR] = 2.21, 95% confidence interval [CI]: 1.67–2.91, p < 0.001). Notably, among women, the corresponding association was even stronger, with the CI+/LMM+ group exhibiting a higher mortality risk (HR = 3.03, 95% CI: 2.26–4.05, p < 0.001). Age-stratified analyses showed that among participants aged <80 years, those in the CI+/LMM+ group had the highest risk of all-cause mortality (HR = 3.66, 95% CI: 2.22–6.04, p < 0.001). Similarly, among participants aged ≥80 years, the CI+/LMM+ group remained significantly associated with an elevated mortality risk (HR = 2.04, 95% CI: 1.64–2.54, p < 0.001). Overall, these findings indicate that the coexistence of cognitive impairment and low muscle mass is associated with the highest risk of all-cause mortality across both sex and age subgroups, with more pronounced associations observed among women and individuals younger than 80 years.

**Table 4. Risks of all-cause mortality according to the presence of cognitive impairment and low muscle mass, stratified by gender and age.**

| Variable | HR (95%CI) | P value | P for interaction |
|---|---|---|---|
| Gender | | | 0.004 |
| Male[a] | | | |
| CI-/LMM- | 1 (ref) | | |
| CI-/LMM+ | 1.27 (1.03, 1.56) | 0.025 | |
| CI+/LMM- | 2.03 (1.47, 2.80) | <0.001 | |
| CI+/LMM+ | 2.21 (1.67, 2.91) | <0.001 | |
| Female[a] | | | |
| CI-/LMM- | 1 (ref) | | |
| CI-/LMM+ | 1.50 (1.14, 1.97) | <0.001 | |
| CI+/LMM- | 1.96 (1.31, 2.92) | <0.001 | |
| CI+/LMM+ | 3.03 (2.26,4.05) | <0.001 | |
| Age,years | | | <0.001 |
| <80 years[b] | | | |
| CI-/LMM- | 1 (ref) | | |
| CI-/LMM+ | 1.24 (0.94, 1.64) | 0.119 | |
| CI+/LMM- | 3.26 (2.09, 5.08) | <0.001 | |
| CI+/LMM+ | 3.66 (2.22, 6.04) | <0.001 | |
| ≥80 years[b] | | | |
| CI-/LMM- | 1 (ref) | | |
| CI-/LMM+ | 1.17 (0.95, 1.43) | 0.123 | |
| CI+/LMM- | 1.42 (1.07, 1.89) | 0.016 | |
| CI+/LMM+ | 2.04 (1.64, 2.54) | <0.001 | |

Model[a]: adjusted for age, residence, marriage, education, average household income, physical exercise, drinking, smoking, BMI, heart disease, hypertension, diabetes, stroke, BADL disability, IADL disability, respiratory diseases, arthritis, cancer, sensory impairment.

Model[b]: adjusted for residence, gender, marriage, education, average household income, physical exercise, drinking, smoking, BMI, heart disease, hypertension, diabetes, stroke, BADL disability, IADL disability, respiratory diseases, arthritis, cancer, sensory impairment.CI, cognitive impairment; LMM, low muscle mass; BMI, Body mass index; HR, hazard ratio; ref, reference.

## Discussion

Cognitive impairment (CI) and low muscle mass (LMM) are highly prevalent among older adults, and their prevalence is expected to increase substantially with the acceleration of global population aging, posing a major public health challenge [21,22]. Using baseline and longitudinal follow-up data from 5,625 participants, this study examined the associations of CI and LMM with all-cause mortality. The results demonstrated a significant association between CI and LMM, and their coexistence was associated with a markedly increased risk of all-cause mortality. In particular, participants in the CI+/LMM+ group exhibited the highest mortality risk. Both unadjusted and fully adjusted models indicated that CI and LMM were independently associated with elevated mortality risk. Sensitivity analyses further confirmed the robustness of these findings. In addition, subgroup analyses revealed that sex and age modified the associations between CI, LMM, and all-cause mortality. Notably, the coexistence of CI and LMM was more strongly associated with mortality risk among women and individuals younger than 80 years.

Consistent with existing literature, this study demonstrated a significant and independent association between cognitive impairment (CI) and low muscle mass (LMM), in line with previous reports [6]. Further analyses showed that CI alone and LMM alone were each associated with an increased risk of all-cause mortality, corroborating earlier studies documenting the independent effects of cognitive impairment and low muscle mass on mortality risk [8,23]. Beyond these established associations, the novelty of the present study lies in its examination of the joint effect of CI and LMM, revealing that their coexistence confers a substantially elevated risk of all-cause mortality. Moreover, relatively few studies have explored

effect heterogeneity by sex and age. By conducting subgroup analyses, this study provides more granular evidence, indicating that women and individuals younger than 80 years experience a particularly heightened mortality risk when both conditions are present. Using data from the China Longitudinal Healthy Longevity Survey, we further observed that 69.9% of participants with cognitive impairment (791/1,132) also had low muscle mass, while 22.1% of those with low muscle mass (791/3,586) also exhibited cognitive impairment. Given the rapidly increasing incidence and prevalence of both conditions, their frequent co-occurrence may represent an important contributor to heightened health risks in aging societies.

To better contextualize these findings, the observed association between cognitive impairment (CI), low muscle mass (LMM), and increased mortality may reflect the influence of several overlapping biological processes proposed in prior literature. Chronic low-grade inflammation, characterized by elevated circulating levels of pro-inflammatory cytokines such as interleukin-6 and tumor necrosis factor-α, has been linked to skeletal muscle protein catabolism as well as neuroinflammatory changes in aging populations, potentially contributing to declines in both physical and cognitive resilience [24–26]. Similarly, oxidative stress and mitochondrial dysfunction have been implicated in age-related muscle loss and cognitive impairment, suggesting a shared vulnerability across multiple organ systems during aging [27–29]. Neuroendocrine and metabolic dysregulation may also play a role. Reduced levels of brain-derived neurotrophic factor (BDNF), which is critical for synaptic plasticity and neuronal survival, have been associated with impaired cognitive function, while alterations in muscle-derived signaling may further limit neurotrophic support [30,31]. In parallel, insulin resistance—commonly observed in individuals with low muscle mass—has been linked to disrupted central insulin signaling and accelerated cognitive decline [32,33]. Moreover, age-related hormonal alterations, including declines in anabolic hormones and dysregulation of growth-related factors, have been associated with both reduced muscle protein synthesis and cognitive impairment in older adults [34,35]. Emerging evidence further suggests that fibroblast growth factor signaling and gut microbiota dysbiosis may influence aging-related metabolic and inflammatory pathways relevant to both muscle and brain health [36–38]. Importantly, these biological processes should be interpreted as plausible explanatory frameworks derived from prior studies, rather than as causal pathways established by the present observational analysis.

In line with the subgroup findings, the observed sex- and age-specific differences may reflect variations in biological susceptibility and overall health status, as suggested by previous studies. Our results indicate that the relationship between the coexistence of cognitive impairment (CI) and low muscle mass (LMM) and all-cause mortality differs significantly by sex and age. Specifically, mortality risk in the CI+/LMM+ group was significantly higher than in the CI−/LMM− group for both men and women, with a more pronounced effect observed among women. In men, the mortality risk in the CI+/LMM+ group was 2.21 times that of the reference group, whereas in women the risk increased to 3.03 times. Previous studies have also reported a stronger association between low muscle mass and cognitive impairment in women [6]. These sex differences may, in part, be related to factors such as brain volume, body composition, inflammatory profiles, and sex hormones. Brain volume has been identified as a key determinant of sex differences in cognitive decline [39], and women tend to exhibit more prominent inflammatory markers associated with skeletal muscle mass decline [40]. The role of sex hormones is also notable, although evidence suggests limited effectiveness of androgen therapy for low muscle mass in men [41]. Sex-specific differences in hippocampal plasticity, activation, and morphology have further been reported, along with distinct distributions of sex hormone receptors in the hippocampus [42]. In women, postmenopausal declines in estrogen levels are closely linked to cognitive decline [43], while endocrine changes such as reduced estrogen and elevated follicle-stimulating hormone (FSH) levels have also been associated with muscle mass loss [44]. Age-related modification of these associations was likewise evident. Among individuals aged <80 years, the CI+/LMM+ group exhibited the highest mortality risk (HR: 3.66), whereas among those aged ≥80 years, although mortality risk remained significantly elevated (HR: 2.04), it was comparatively lower. Previous longitudinal research based on the CLHLS cohort has reported that the association between muscle mass and cognitive function appears to be more pronounced among individuals younger than 80 years, suggesting potential age-related heterogeneity in this relationship [9]. With advancing age, this association may attenuate, potentially due to accumulating systemic inflammation and insulin resistance, which could weaken the muscle–brain metabolic coupling proposed in previous studies [45].

Taken together, this study, based on a large nationally representative longitudinal cohort of older adults in China (CLHLS), demonstrates that the coexistence of cognitive impairment (CI) and low muscle mass (LMM) is associated with a markedly increased risk of all-cause mortality, particularly among women and individuals younger than 80 years. Unlike studies focusing on single conditions or region-specific cohorts, these findings suggest that the concurrent presence of CI and LMM may identify a subgroup of older adults with heightened overall vulnerability, rather than reflecting a disease-specific mechanism. In the context of China's rapidly aging population, recognizing such high-risk profiles may inform population-level risk stratification and support comprehensive geriatric assessment beyond single-condition management. Several limitations should be acknowledged. First, the study population consisted exclusively of older adults in China; therefore, external validation in ethnically and geographically diverse populations is warranted. Second, low muscle mass was assessed using screening-based indicators, without direct measurements of muscle mass, muscle strength, or physical performance. Additionally, cause-specific mortality could not be examined due to database constraints, and residual confounding from unmeasured factors cannot be fully excluded. Future studies incorporating standardized and objective assessments of muscle quantity and function, more refined cognitive phenotyping, and broader outcome measures are needed to further clarify the relationships between CI, LMM, and mortality.

## Conclusion

Based on longitudinal data from the China Longitudinal Healthy Longevity Survey (CLHLS), this study demonstrates that the coexistence of cognitive impairment (CI) and low muscle mass (LMM) is associated with a substantially increased risk of all-cause mortality among older adults in China. Individuals with both conditions exhibited the highest mortality risk, and this association remained robust across multiple sensitivity analyses. Stronger associations observed among women and individuals younger than 80 years suggest that sex and age may modify the relationship between CI, LMM, and mortality. Given that CI and LMM were assessed using screening-based measures rather than clinical diagnoses, these findings should be interpreted as identifying a high-risk population rather than a disease-specific pathway. From a public health perspective, recognizing the combined presence of CI and LMM may aid risk stratification and support comprehensive geriatric assessment in aging populations.

## Supporting information

**S1 File. S1 Table. VIF results of confounders included in the multivariate regression models; Adjusted HRs of CI or LMM status and all-cause mortality, excluding the participants with a previous episode of stroke; Adjusted HRs of CI or LMM status and all-cause mortality, excluding the participants with a previous episode of heart disease; Adjusted HRs of CI or LMM status and all-cause mortality, excluding participants who had died within 2 years of follow-up.**
(DOCX)

## Acknowledgments

The authors would like to thank all co-workers and the reviewers. We also gratefully acknowledge the team of the Chinese Longitudinal Healthy Longevity Survey (CLHLS) for providing access to the data and training on its use.

## Author contributions

**Conceptualization:** Jiefen Ou, Huixia Cao.

**Data curation:** Huixia Cao.

**Formal analysis:** Jiefen Ou, Huixia Cao.

**Investigation:** Jiefen Ou, Huixia Cao.

**Methodology:** Jiefen Ou, Huixia Cao.

**Software:** Jiefen Ou, Huixia Cao.

**Supervision:** Manzhi Gao.

**Validation:** Yingying Zhong.

**Writing – original draft:** Jiefen Ou, Huixia Cao, Yingying Zhong.

**Writing – review & editing:** Manzhi Gao.

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
