## [Decision Letter · Decision Letter 0]

29 Oct 2025

Dear Dr. Ou,

 We have received two good review reports and on their grounds I decided to give you a change for truely major revision. Although the paper is well-written and presents an interesting analysis of secondary data, the reviewers have raised serious doubts on crucial aspects of your paper, including the definition and proxy-meausrement of (possible?) sarcopenia. For the benefit of a possible publication, you really have to deal with each and every comment, and your major revision will then again be reviewed seriously.

We look forward to receiving your revised manuscript.

Kind regards,

Robbert Huijsman, PhD

Academic Editor

PLOS ONE

Journal Requirements:

2. Please note that your Data Availability Statement is currently missing the repository name. If your manuscript is accepted for publication, you will be asked to provide these details on a very short timeline. We therefore suggest that you provide this information now, though we will not hold up the peer review process if you are unable.

Reviewers' comments:

Reviewer's Responses to Questions

**Comments to the Author**

1. Is the manuscript technically sound, and do the data support the conclusions?

Reviewer #1: No

Reviewer #2: Partly

2. Has the statistical analysis been performed appropriately and rigorously?

Reviewer #1: Yes

Reviewer #2: Yes

3. Have the authors made all data underlying the findings in their manuscript fully available?

Reviewer #1: Yes

Reviewer #2: No

4. Is the manuscript presented in an intelligible fashion and written in standard English?

Reviewer #1: Yes

Reviewer #2: Yes

Reviewer #1: Your manuscript is clear and well-written. However, although the results are interesting, there is a flaw in the design regarding measurement of sarcopenia. The definition of sarcopenia used in your study is not consistent with that recognized by the Asian Working Group of Sarcopenia that you cited as the Asian Obesity Guidelines. The AWGS is specific that loss of strength must accompany loss of muscle (“age-related loss of skeletal muscle mass plus loss of muscle strength and/or reduced physical performance”). Furthermore, possible or potential sarcopenia is not a clearly defined or measurable variable. Either participants are sarcopenic or they are not, just as they are cognitively impaired or they are not. Although you used a measurement strategy based on calf circumference that incorporates BMI (height/weight) that is validated for prediction of appendicular skeletal muscle, it is not, in and of itself, a valid measure of sarcopenia. I note that the Ren et al. (2023) study that used the formula for calculating appendicular skeletal muscle (ASM) referred to it only as low muscle mass. That might be your best approach for describing your study variable. In any case, without measurement of strength (or physical performance as an alternative), you cannot describe your study as evaluating sarcopenia, and you cannot with integrity as a researcher get around this by using “possible” or “potential.”

Reviewer #2: The main problem with this manuscript is that the authors make a quick translation between cognitive impairment and sarcopenia in which they choose the Mini Mental State Examination (MMSE) as a proxy for Cognitive Impairment (CI) and calf circumference (CC) as a proxy for sarcopenia in their introduction. Then they make the step that CI is due Alzheimer’s pathology and low CC is due to sarcopenia and combine these two conditions and arrive at a common pathophysiological mechanism. However a low MMSE (<24) is not the same as Alzheimer’s disease (AD) and reflects multiple possible causes (e.g. level of education, age, depression, drugs cerebrovascular disease etc.) of CI.

The authors further state that calf circumference (CC) is a good proxy for sarcopenia, but a low CC can also be caused by other conditions, e.g. malnutrition, immobility due to neurologic or vascular problems, consequence of sedentary behavior, osteoarthritis etc.

Due to these different causes of CI and low CC it simply not connect these measures with Alzheimer ‘s disease or CI and possible sarcopenia and a common pathway as the authors state.

The researchers are studying the relationship between the impact of an MMSE <24, low CC, and the presence of the combination on mortality using the Kaplan-Meier curve and Hazard Ratio, and then examining the impact of several potential confounders. However, this approach fails to account for several very important confounders, such as ADL dependency, the presence of sensory impairments, osteoarthritis, cancer, lung disease, etc. This can introduce a significant risk of bias.

The inclusion and the exclusion criteria are described on page 4. Patients with a history of memory disorders or illnesses are excluded ( totals 608) while the study included 1,596 people with a minimum MMSE <24. It is not clear what kind of people with a history of memory problems are excluded and why people with an MMSE <24 without an official diagnosis are included. For the group not included, it is unclear whether a diagnosis exists or has not yet been made and is known that many peoples in the community have not yet been diagnosed. Excluding 608 patients in this way could be a bias. Moreover, the authors use only one strict cutoff value of the MMSE, while their database contains data on age distribution and education duration, which would allow them to adjust the MMSE for age and education duration, which could impact the normal values used.

On page 6, the authors discuss calf circumference and the calculation of muscle mass based on a formula. However, the method of calf measurement is not described throughout the study. Moreover, calf circumference appears to be a poor screening method for sarcopenia. How do the authors explain this use?

On page 6, under the study outcomes, Data on the number of patients lost to follow-up are missing. This is, of course, also relevant to the results.

In 2.5, assessment of potential confounders (page 7), it's striking that some data are very detailed, while other, more confounders that are very relevant to this study are only classified very broadly. For example, how is physical activity defined? Moreover, conditions such as heart disease and stroke are not defined, making this a very heterogeneous group. Other variables, such as the presence of malignancy, pulmonary disease, osteoarthritis, and functional level (particularly care dependency), visual and hearing problems, are important confounders that are not described at all, potentially leading to significant bias.

The statistical analyses were performed correctly.

In the discussion starting on page 11, particularly page 12 and following, the authors address the impact of cognitive impairment and potential sarcopenia on all-cause mortality. However, given previous comments, such as the inability to correct for important confounders other than those mentioned in this study, this discussion is flawed in some respects. Certainly, the pathophysiological mechanisms themselves may be correct. But in the present CI is probable much broader than Alzheimer’s disease en represents a heterogeneous group of cognitive disorders with a very heterogeneous etiology. The same is probable true for a low calf circumference and based on different causes.

The conclusion, as outlined on pages 15 and 16, is that these findings should prompt screening for cognitive disorders and probable sarcopenia, so that a multi-target collaborative therapeutic strategy can be developed that addresses both pathways. However, this assertion cannot be made based on the current findings in this study.

Furthermore there is in the manuscript no consequent of terminology: potential and possible sarcopenia are mixed up.

**Do you want your identity to be public for this peer review?** For information about this choice, including consent withdrawal, please see our Privacy Policy

Reviewer #1: No

Reviewer #2: No

---

## [Author Response · Author response to Decision Letter 1]

6 Jan 2026

Dear Editors and Reviewers,

We would like to express our sincere appreciation for your letter and the valuable comments provided on our manuscript, entitled “The impact of cognitive impairment and low muscle mass on all-cause mortality among older adults in China: an empirical analysis based on CLHLS cohort data” (Manuscript ID: PONE-D-25-45768). These comments have been highly valuable and extremely helpful in revising and improving our manuscript, as well as providing important guidance for our research. We have carefully studied all the comments and have undertaken a substantial revision of the manuscript, including revisions to the core conceptual framework, terminology, measurement interpretation, confounder adjustment, and the scope of inference, in order to fully address the reviewers’ major concerns. We hope that the revisions and responses presented below meet with your approval, and the main corrections in the manuscript, together with our point-by-point responses to the Editors’ and reviewers’ comments, are provided as follows.

Responses to the Editors

1.Editorial Corrections:

Comment #1: The reviewers raised serious concerns regarding the definition and proxy measurement of (possible) sarcopenia. The authors must carefully address each comment, particularly regarding conceptual clarity and methodological rigor.

Response: We sincerely thank the Editors for this important guidance. We fully acknowledge that the terminology and conceptual framing related to sarcopenia in the original submission required substantial revision. In response, we undertook a comprehensive restructuring of the manuscript. Specifically, we no longer describe or interpret the exposure variable as sarcopenia, nor as “possible” or “potential” sarcopenia, and instead consistently use the term low muscle mass (LMM) throughout the manuscript, clearly distinguishing it from clinically diagnosed sarcopenia. We explicitly clarify that calf circumference–based estimation was applied solely for epidemiological risk stratification rather than for clinical diagnosis, and we acknowledge that muscle strength and physical performance data were not available in the dataset. (Assessment of low muscle mass See: Page 7, Lines 122-142)

Responses to Reviewer 1

Comment #1: The definition of sarcopenia used in the study is not consistent with the Asian Working Group for Sarcopenia. Without muscle strength or physical performance, the authors cannot describe the condition as sarcopenia. The use of “possible” or “potential” sarcopenia is not acceptable.

Response: We sincerely thank the reviewer for this critical and important comment. We fully agree with the reviewer’s assessment. In response, we made several fundamental revisions to the manuscript. All references to sarcopenia, possible sarcopenia, or potential sarcopenia were completely removed from the title, abstract, keywords, tables, figures, and main text. The exposure variable is now consistently defined as low muscle mass (LMM), estimated using a validated anthropometric prediction equation. Consistent with Ren et al. (2023), we explicitly restrict our interpretation to low muscle mass rather than sarcopenia, and we do not equate anthropometric estimates with clinical diagnosis. We explicitly clarify that, due to the absence of muscle strength and physical performance data, this study does not assess sarcopenia as defined by the AWGS and does not imply any clinical diagnosis. Accordingly, the Methods section was revised to emphasize that the measurement approach was intended for population-based epidemiological analysis rather than clinical classification (Methods, “Assessment of low muscle mass”). In addition, the Discussion and Conclusions were revised to ensure that all interpretations are strictly limited to low muscle mass as a risk marker, rather than a diagnostic entity.These revisions were implemented consistently throughout the manuscript, including the Title, Abstract, Tables, Figures, Introduction, Methods (Assessment of low muscle mass, See: Page 7, Lines 122-142), Discussion, and Conclusion.

Responses to Reviewer #2

Comment #1: MMSE <24 is not equivalent to Alzheimer’s disease, and cognitive impairment has multiple etiologies. Similarly, low calf circumference may result from many conditions other than sarcopenia. The manuscript overstates common pathophysiological mechanisms.

Response: We thank the reviewer for highlighting this important issue. We fully agree. In the revised manuscript, we removed all language implying that cognitive impairment measured by the MMSE represents Alzheimer’s disease or any specific neurodegenerative pathology. We explicitly clarify that cognitive impairment assessed by the MMSE is a heterogeneous condition with multiple potential causes, including educational attainment, age, vascular disease, depression, and other factors (Introduction; Discussion). Accordingly, all mechanistic discussions were revised to frame biological pathways as plausible hypotheses based on prior literature rather than as direct explanations supported by our data. We also clearly state that the observed associations reflect epidemiological risk patterns rather than disease-specific mechanisms. Relevant revisions appear in the Introduction and Discussion sections. (Introduction See: Page 3, Lines 48-53; Discussion, See: Page 20-21, Lines 346-366)

Comment #2: The reviewer is concerned that several important comorbidities, such as heart disease and stroke, are not clearly defined. Without specifying how these conditions were assessed, these categories may represent highly heterogeneous groups, potentially introducing bias into the analyses.

Response: We thank the reviewer for this important comment. In the revised manuscript, we have clarified the operational definitions of heart disease and stroke conditions. Heart disease was defined based on self-reported physician diagnoses, including coronary heart disease, myocardial infarction, heart valve disease, and other cardiac conditions, as collected through standardized CLHLS survey items. Stroke was similarly defined as self-reported physician-diagnosed acute cerebrovascular events, including cerebral infarction and cerebral hemorrhage. (See: Page 8-9, Lines 165-168)

Comment #3: The reviewer notes that physical activity is an important confounder in this study, yet its definition and measurement are unclear. It is not specified how physical activity was assessed or categorized, which limits the interpretability of the adjusted models.

Response: We thank the reviewer for this important comment. In response, we have revised the manuscript to improve clarity and precision by replacing the term physical activity with physical exercise throughout the Methods and statistical models. Physical exercise was assessed using a standardized CLHLS survey item asking participants, “Do you take exercise regularly at present?” and was coded dichotomously as yes or no. This variable provides a simplified, self-reported indicator of regular exercise behavior among community-dwelling older adults and has been used in prior analyses based on the CLHLS dataset. (e.g., Association between nut consumption and mortality among Chinese older people, CLHLS 2008–2018) (See: Page 8, Lines 162-163)

Comment #4: Important confounders such as ADL dependency, sensory impairment, cancer, lung disease, and osteoarthritis were not adequately accounted for, leading to potential bias.

Response: We sincerely appreciate this insightful comment. To address this concern, we substantially expanded the set of covariates included in the fully adjusted Cox models. Specifically, we added measures of functional status (BADL and IADL impairment), sensory impairment (vision or hearing), respiratory diseases, arthritis, and cancer. These variables are now incorporated into the fully adjusted model (Model 4) and are described in detail in the Methods section (“Assessment of potential confounders”). This revision substantially reduces potential residual confounding and strengthens the robustness of our findings. (See: Page 9, Lines 170-174)

Comment #5: The reviewer raises concerns that functional level, particularly care dependency, was not adequately addressed. Although some functional measures were included, it is unclear whether these variables sufficiently capture dependency or care needs among older adults.

Response: We thank the reviewer for highlighting this issue. To better capture functional status and dependency, we incorporated both Basic Activities of Daily Living (BADL) and Instrumental Activities of Daily Living (IADL) into the fully adjusted models. BADL impairment was defined as the inability to independently perform at least one basic self-care activity, while IADL limitation reflected difficulties in performing more complex activities required for independent living. In this study, BADL and IADL impairments were used as proxies for functional dependence and care needs among community-dwelling older adults, which is a common approach in large epidemiological studies. (e.g., Functional disability in basic and instrumental activities of daily living among older adults globally: a systematic review and meta-analysis) (See: Page 9, Lines 174-178)

Comment #6: Concerns regarding inclusion and exclusion criteria for participants with memory disorders, MMSE cutoff values, and lack of education adjustment.

Response: We thank the reviewer for this important methodological comment. In the revised manuscript, we clarify that participants with a diagnosed dementia were excluded, whereas individuals with low MMSE scores in the absence of a formal diagnosis were retained, reflecting real-world community-based cognitive screening practices. We also adopted education-specific MMSE cutoff values, consistent with established approaches in Chinese aging research, to better account for educational heterogeneity. The rationale for these methodological decisions is now clearly detailed in the Methods section (“Assessment of cognitive impairment”). (Study design and population See: Page 5, Lines 98-100; Assessment of cognitive impairment, See: Page 6-7, Lines 117-121)

Comment #7: Calf circumference measurement methods were not clearly described, and CC is a poor screening tool for sarcopenia.

Response: We agree with the reviewer’s concern and have revised the manuscript accordingly. We now explicitly state that the participant was asked to put the sole of his or her right foot onto the ground and to make his or her right calf and right thigh into a 90-degree angle while seated. A measuring tape was used to measure the circumference of the right calf at its widest point. Calf circumference was rounded to the nearest integer. We further clarify that calf circumference–based estimation was used solely to estimate muscle mass for population-based epidemiological analyses and not for the diagnosis of sarcopenia. These clarifications have been incorporated into the Methods section (“Assessment of low muscle mass”). (See: Page 7, Lines 129-136)

Comment #8: Loss to follow-up data were missing.

Response: We thank the reviewer for this observation. In the revised manuscript, we have added explicit information on loss to follow-up, including the number of participants lost and the censoring strategy, in the “Study outcomes” section. (See: Page 8, Lines 151-152)

Comment #9: The discussion and conclusions overstate implications for screening and intervention.

Response: We fully agree and appreciate this important comment. Accordingly, we revised the Discussion to avoid overinterpretation and causal language and reframed the Conclusions to emphasize risk stratification and population-level vulnerability rather than clinical screening or intervention recommendations. We also clearly state that the findings should be interpreted as identifying high-risk groups, not as evidence for disease-specific mechanisms or treatment strategies. These revisions are reflected throughout the Discussion and Conclusion sections. (Discussion, See: Page 19-23, Lines 318-409; Conclusion, See: Page 23, Lines 410-421)

Comment #10: Inconsistent use of “possible” and “potential” sarcopenia terminology.

Response: We thank the reviewer for pointing this out. As noted above, all such terminology has been completely removed, and the manuscript now consistently uses low muscle mass (LMM) throughout.

We express our sincere gratitude to the reviewers and the editorial team for your valuable guidance and support throughout this process. Thank you for considering our revised submission for reconsideration. We look forward to receiving your final decision.

Sincerely,

Jiefen Ou

Corresponding author: Jiefen Ou

Email: Ojf2854529676@163.com

---

## [Decision Letter · Decision Letter 1]

2 Feb 2026

Dear Dr. Ou,

Thank you for submitting the revision of your manuscript to PLOS ONE. After careful consideration, we feel that it has merit but does not fully meet PLOS ONE’s publication criteria as it currently stands.  After your good major revisions, there still remain some minor points, as show by the reviewer. Please follow these comments and suggestions one by one for a minor second revision. Therefore, we invite you to submit a revised version of the manuscript.

We look forward to receiving your revised manuscript.

Kind regards,

Robbert Huijsman, PhD

Academic Editor

PLOS One

Journal Requirements:

Reviewers' comments:

Reviewer's Responses to Questions

**Comments to the Author**

Reviewer #1: (No Response)

2. Is the manuscript technically sound, and do the data support the conclusions?

Reviewer #1: Yes

3. Has the statistical analysis been performed appropriately and rigorously?

Reviewer #1: Yes

4. Have the authors made all data underlying the findings in their manuscript fully available?

Reviewer #1: Yes

5. Is the manuscript presented in an intelligible fashion and written in standard English?

Reviewer #1: Yes

Reviewer #1: Thank you for the opportunity to review your manuscript. I sincerely appreciate the extensive revisions you made and have just a few remaining questions and concerns.

Introduction

Lines 55-57 – Although an excellent paper, Reference #4 does not provide support for you statement that low muscle is highly prevalent, etc. Please review it as needed and provide appropriate evidence for these two sentences.

Line 70 – I appreciate your point in this sentence but 1.57-fold isn’t an appropriate term to describe the data. “Fold” is not typically used until risk is doubled (2.0), when “two-fold” would be an appropriate term. The data (1.57) mean a 57% increase in risk, which is likely the best way to describe it. Your point is still well-taken and convincing.

Line 73 – Please cite at least one population-based observational study to support your point.

Methods

Line 96 – Reference #10 appears to be incomplete. Please check it.

Figure 1 and Line 153 – You report that 110 participants were lost to follow up but I don’t see that in Figure 1. Were they excluded prior to determining the final sample (N = 5625) or afterward? Please clarify.

Lines 119-121 and Table 1 – Your description of education used to define cognitive impairment cutoff points is very interesting but it’s not clear how the criteria in text (Lines 119-121) align with the variables in Table 1. For example, would less than 6 years of schooling be considered illiterate? Perhaps a brief explanation to assist interested readers is needed here.

Lines 139-143 – The source of your prediction equation isn’t clear. It could not have been your Reference #15, because the editorial by Bahat did not report any prediction equations. Please check your source in this case.

Results

Your analysis is detailed and thorough. The trends you identified are most interesting.

There seems to be a minor errors in your Subgroup Analysis. You report that the CI+/LMM+ group had the highest mortality risk (HR 3.26) for those under 80 years of age. However, the data in Table 4 indicate that actually, the CI+/LMM- group had the highest risk (HR 3.66). Please make whatever correction is needed so that your report is consistent with your data.

Discussion

Your discussion is well thought out and you make some intriguing points. Well done.

Lines 389-391 – Your point about consistent findings is a bit simplistic when presumably the same sample was used for analysis (CLHLS cohort). Wouldn’t it be intuitive that the associations would hold true? Perhaps you could find another population-based study with a different sample to make your comparison? It would certainly strengthen your point considerably.

**Do you want your identity to be public for this peer review?** For information about this choice, including consent withdrawal, please see our Privacy Policy

Reviewer #1: No

---

## [Author Response · Author response to Decision Letter 2]

6 Feb 2026

Dear Editors and Reviewers,

We are profoundly grateful for the opportunity to resubmit our work. We wish to thank the Editor for their time and the reviewers for providing such meticulous and valuable feedback. Your professional expertise and rigorous attitude have inspired our team significantly, and we feel that our research has been greatly enhanced by your guidance. We sincerely apologize for any shortcomings in the previous version and are deeply moved by the time you spent helping us refine our study. We have revised the paper to the best of our ability and remain open to any further suggestions you may have.

Responses to Reviewer 1

Comment #1: Although an excellent paper, Reference #4 does not provide support for the statement that low muscle mass is highly prevalent. Please provide appropriate evidence.

Response: We thank the reviewer for this important observation. We carefully re-evaluated Reference #4 and agree that it does not directly support the statement regarding the high prevalence of low muscle mass. We have therefore revised the relevant sentences and added population-based epidemiological evidence demonstrating the prevalence of low muscle mass among older adults. Specifically, we incorporated recent large-scale studies based on the CLHLS and other population-based cohorts to provide appropriate empirical support (revised Introduction, Lines 53–55, Reference: Associations of Chinese-modified MIND diet with low muscle mass and physical performance among old adults in China: findings from the CLHLS 2018 national survey. )

Comment #2: The term “1.57-fold” is not appropriate; a 57% increase in risk would be more accurate.

Response: We appreciate this helpful clarification. We have revised the wording accordingly, replacing “1.57-fold” with “a 57% increase in the risk of all-cause mortality,” which more accurately reflects the magnitude of the association (revised Introduction, Line 70)

Comment #3: Please cite at least one population-based observational study to support this point.

Response: We agree with the reviewer’s suggestion. We have added a population-based observational study examining the association between cognitive impairment and low muscle mass in older adults to support this statement. The corresponding citation has been included in the revised Introduction (Lines 72–76, Reference: The relationship between skeletal muscle mass and cognitive impairment in older adults: a longitudinal study based on CLHLS.)

Comment #4: Reference #10 appears to be incomplete.

Response: Thank you for pointing this out. We have carefully checked Reference #10 and corrected the incomplete information. The reference list has been updated to include the full bibliographic details (Lines 476–478, Reference: 12. Yi Z. Introduction to the Chinese Longitudinal Healthy Longevity Survey (CLHLS). In: Yi Z, Poston DL, Vlosky DA, Gu D, editors. Healthy Longevity in China: Demographic, Socioeconomic, and Psychological Dimensions. Dordrecht: Springer; 2008. p. 23–38.)

Comment #5: You report that 110 participants were lost to follow-up, but this is not clear in Figure 1. Please clarify.

Response: We appreciate this comment. We have clarified the handling of participants lost to follow-up in both the figure and the text. Specifically, Figure 1 has been revised to explicitly indicate “Lost to follow-up (n = 110),” and the Methods section now clearly states that these participants were excluded prior to the formation of the final analytical sample (revised Methods, Lines 108; Lines 154-157)

Comment #6: The relationship between education categories used for cognitive impairment cutoffs and those presented in Table 1 is unclear.

Response: Thank you for highlighting this issue. To improve clarity, we have added a brief explanatory sentence in the Methods section specifying how educational categories correspond across the manuscript. We now explicitly state that “illiterate” refers to 0 years of schooling, “primary school” corresponds to 1–6 years of schooling, and “secondary school or above” refers to more than 6 years of schooling. This clarification aligns the education-specific MMSE cutoff points with the categories shown in Table 1 (revised Methods, Lines 118–122; Lines 164–165)

Comment #7: The source of the prediction equation is unclear, and Reference #15 does not report any prediction equations.

Response: We thank the reviewer for identifying this error. We carefully rechecked the source of the appendicular skeletal muscle mass prediction equation and confirmed that it was incorrectly attributed. We have now corrected the citation by replacing Reference #15 with the original study that developed and validated the anthropometric prediction equation. The corresponding text and reference list have been revised accordingly (revised Methods, Lines 128–131)

Comment #8: There is an inconsistency between the text and Table 4 regarding which group has the highest mortality risk among participants under 80 years of age.

Response: We apologize for this oversight and thank the reviewer for bringing it to our attention. We have corrected the text in the Results section to ensure consistency with Table 4. The revised text now accurately states that among participants aged <80 years, the CI+/LMM+ group exhibited the highest mortality risk (HR = 3.66). All related descriptions have been carefully checked for consistency (revised Results and Table 4, Lines 307–309)

Comment #9: The statement regarding consistent findings may be simplistic, as the same cohort (CLHLS) was used. Consider comparison with other populations.

Response: We thank the reviewer for this thoughtful comment. We agree that the age-specific finding should be interpreted with appropriate caution. In the revised Discussion, we have clarified that the evidence regarding stronger associations among individuals younger than 80 years is derived from previous longitudinal analyses conducted within the same CLHLS cohort, rather than from independent external populations. Accordingly, we have framed this observation as supportive background evidence suggesting potential age-related heterogeneity, rather than as a universally established finding. Our study therefore serves as an independent longitudinal validation of this age-related pattern within a large nationally representative cohort (revised Discussion, Lines394–397)

We are deeply grateful to the reviewers and the editorial team for the insightful guidance provided throughout the review process. Your expertise has been instrumental in enhancing the quality of our work. We sincerely appreciate your continued consideration of our revised manuscript and look forward to your feedback.

Sincerely,

Jiefen Ou

Corresponding author: Jiefen Ou

Email: Ojf2854529676@163.com

---

## [Editor Report · Decision Letter 2]

9 Feb 2026

The impact of cognitive impairment and low muscle mass on all-cause mortality among older adults in China: an empirical analysis based on CLHLS cohort data

PONE-D-25-45768R2

Dear Dr. Ou,

We’re pleased to inform you that your manuscript has been judged scientifically suitable for publication and will be formally accepted for publication once it meets all outstanding technical requirements.

Kind regards,

Robbert Huijsman, PhD

Academic Editor

PLOS One
---

## [Editor Report · Acceptance letter]

PONE-D-25-45768R2

PLOS One

Dear Dr. Ou,

I'm pleased to inform you that your manuscript has been deemed suitable for publication in PLOS One. Congratulations! Your manuscript is now being handed over to our production team.

Kind regards,

on behalf of

Professor Robbert Huijsman

Academic Editor

PLOS One